# Analysis of CircRNA Expression in Peripheral Blood of Holstein Cows in Response to Heat Stress

**DOI:** 10.3390/ijms241210150

**Published:** 2023-06-15

**Authors:** Congcong Zhang, Shuhui Wang, Lirong Hu, Hao Fang, Gong Chen, Xiaojuan Ma, Ying Yu, Yachun Wang, Qing Xu

**Affiliations:** 1College of Life Sciences and Bioengineering, Beijing Jiaotong University, Beijing 100044, China; 21121613@bjtu.edu.cn (C.Z.); shuhui.wang@dauphine.eu (S.W.); 18121612@bjtu.edu.cn (H.F.); 22121620@bjtu.edu.cn (G.C.); 18636764982m@gmail.com (X.M.); 2National Engineering Laboratory for Animal Breeding, Key Laboratory of Animal Genetics, Breeding and Reproduction, MARA, College of Animal Sciences and Technology, China Agricultural University, Beijing 100193, China; b20193040324@cau.edu.cn (L.H.); yuying@cau.edu.cn (Y.Y.)

**Keywords:** dairy cows, heat stress, circRNA, RNA-seq

## Abstract

The present study aimed to identify key circRNAs and pathways associated with heat stress in blood samples of Holstein cows, which will provide new insights into the molecular mechanisms driving heat stress in cows. Hence, we evaluated changes in milk yield, rectal temperature, and respiratory rate of experimental cows between heat stress (summer) and non-heat stress (spring) conditions with two comparisons, including Sum1 vs. Spr1 (same lactation stage, different individuals, 15 cows per group) and Sum1 vs. Spr2 (same individual, different lactation stages, 15 cows per group). Compared to both Spr1 and Spr2, cows in the Sum1 group had a significantly lower milk yield, while rectal temperature and respiratory rate were significantly higher (*p* < 0.05), indicating that cows in the Sum1 group were experiencing heat stress. In each group, five animals were chosen randomly to undergo RNA-seq. The results reveal that 140 and 205 differentially expressed (DE) circRNAs were screened in the first and second comparisons, respectively. According to the gene ontology (GO) and Kyoto Encyclopedia of Genes and Genomes (KEGG) pathway analysis, these DE circRNAs were mainly enriched in five signaling pathways, including choline metabolism, the PI3K/AKT signaling pathway, the HIF-1 signaling pathway, the longevity-regulating pathway, and autophagy. Then, we obtained the top 10 hub source genes of circRNAs according to the protein–protein interaction networks. Among them, ciRNA1282 (*HIF1A*), circRNA4205 (*NR3C1*), and circRNA12923 (*ROCK1*) were enriched in multiple pathways and identified as binding multiple miRNAs. These key circRNAs may play an important role in the heat stress responses of dairy cows. These results provide valuable information on the involvement of key circRNAs and their expression pattern in the heat stress response of cows.

## 1. Introduction

With the global temperature rise, the dairy industry is confronted with a significant challenge brought about by severely damaging impacts of heat stress. Heat stress increases mortality rates and threatens the health of livestock, especially in highly productive and intolerant species. The increase in ambient temperature–humidity index (THI) in summer will affect the balance of organisms and lead to heat stress in animals. Dairy cows are more sensitive to high THI due to their high metabolism and low heat dissipation. Heat stress in dairy cows has become increasingly severe, causing huge economic losses to the dairy industry. It has been reported that heat stress has caused a loss to the dairy industry of $1.5 billion each year [1]. Therefore, it is necessary to develop strategies to reduce the damage caused by heat stress in animal husbandry. Investigating the effect of heat stress on cow performance and its potential mechanisms is very important to fundamentally reduce the negative effect of heat stress. With the rapid development of sequencing technology, genome-wide association studies (GWAS) [2] and transcriptome [3], metabolome [4,5], and microbiome [6,7] analyses were used to screen biochemical indicators and identify molecular mechanisms associated with heat stress in dairy cows. However, there are few studies about the regulation of post-transcriptional noncoding RNAs (ncRNAs), such as long noncoding RNAs (lncRNA), microRNAs (miRNAs), and circular RNAs (circRNAs), in dairy cows after exposure to heat stress conditions, especially studies focusing on circRNAs.

Recently, circular RNA (circRNA) has attracted significant attention in research due to its potentially rich functions in cells. CircRNAs are a large class of ncRNAs ubiquitously expressed in eukaryotic cells. They have a covalently bonded closed-loop structure, resulting in their stronger stability than linear RNAs [8]. Subsequent research has tried to uncover the functional role of circRNAs in various species and tissues. A handful of circRNAs have been demonstrated to have functions in post-transcriptional regulation, acting as miRNA sponges [9]. An increasing number of studies have been conducted to improve livestock milk production [10], meat quality [11] and skeletal muscle development [12] by describing the mechanisms of action and related pathways of circRNAs. Hao et al. [10] found 18 classes of miRNAs play essential roles in the regulation of lactation and mammary growth through targeting circRNAs in sheep. CircRNA is also associated with the development of livestock embryos, and circTUT7 regulates the expression of HMG20B in a ceRNA mechanism, suggesting their critical functions in embryonic skeletal muscle development in porcine embryonic muscle [13]. Regarding the role of circRNAs in meat quality in cows, circFUT10 was observed to bind miR-133a to regulate myoblast differentiation in bovine skeletal muscle tissue [14]. However, little is known about the role of circular RNAs in the responses of dairy cows undergoing heat stress.

Limited studies on circRNA and heat stress in dairy cows have been reported. Wang et al. [15,16] identified several circRNA-associated ceRNA networks involved in milk fat metabolism under heat stress and found circEZH2 regulated milk fat metabolism through miR-378b sponge activity. Furthermore, copious changed circRNAs were discovered in cow mammary gland tissues [15,17] and hypothalamic and pituitary tissues [18] under heat stress. Among all these papers, different tissue samples of cattle under heat stress were used, but not peripheral blood. It is worth mentioning that other cattle tissue samples are more difficult to obtain than peripheral blood, since it is readily available from cows. Moreover, sampling of peripheral blood is less invasive, more convenient, and safer. Especially as one common cellular model, the role and mechanism of cattle peripheral blood mononuclear cells (PBMCs) have been extensively studied [19,20]. In addition, when designing experiments, we had to consider the effect of various factors on the incidence of heat stress, such as individual genetic background, lactation, and breed. Most studies on heat-stress-related circRNAs have selected cows with the same lactation stage as samples, which may ignore the individual differences [15,17]. Liu et al. [21] described 86 expression patterns of mRNA and miRNA in the blood of the same individual cows under different conditions from heat stress to thermal equilibrium. They suggested individual influence on cows’ response to heat stress. Thereby, we investigated the effect of heat stress on circRNA by two experimental comparisons, the same individuals under different lactation stages in comparison1 (Spr1 and Sum1 groups) and different individuals under the same lactation stage in comparison2 (Spr2 and Sum1 groups).

Here, we examined the changes in milk yield, rectal temperature, and respiratory rate of dairy cows under heat stress. Then, we focused on the effect of heat stress on the key biological processes and candidate circRNAs in blood samples of cows through two experimental comparisons. The obtained results comprehensively elucidate the underlying circRNAs of heat stress response and provide a new insight into breeding of thermotolerant dairy cows.

## 2. Results

### 2.1. The Performances of Holstein Cattle under Heat Stress

The experiments were conducted in April and July 2017. Between the Spr (April) and Sum (July) groups, there was obvious difference in the temperature–humidity index (THI) calculated from ambient temperature and humidity (*p* < 0.05). The average THI in spring was about 55, while it increased to about 81 in summer (Table 1). From THI, it can be seen that no heat stress occurred in 2 Spr groups in April, when an average THI only reached to 55. In contrast, the Sum1 group suffered moderate heat stress during July, since the THI reached a high value: 81. Additionally, from Table 1, it is evident that the seven-day average milk yield (7AMY) and milk yield (MY) were generally higher in the Spr1 group than in the Sum1 group (*p* < 0.05). Furthermore, both physiological indicators, rectal temperature (RT) and respiration rate (RR) of cow, were significantly increased in the Sum1 group compared with the Spr2 group.

### 2.2. Identification and Characterization of circRNAs in Holstein Cattle

In order to profile the expression of circRNAs in Holstein cattle, 15 samples from Spr1, Spr2, and Sum1 groups were used to perform RNA-seq and extract the characteristics of circRNAs. After removing the de-duplicated reads, we obtained an average of 74,665,172 reads in each sample, of which 91.45–93.96% of the reads were successfully mapped to the bovine reference genome, and 68.68–78.32% were uniquely mapped to the genome (Appendix A). These results show our sequencing data had good quality, which supported its usage for the subsequent analysis. After removing the linear RNA and ribosomal RNA, 24,250, 26,201, and 24,848 circRNAs were identified in blood samples of Holstein cows from Spr1, Spr2, and Sum1 groups, respectively (Figure 1a). The majority of circRNAs were composed of exons (average = 47.93%) and introns (average = 44.32%), while a small proportion of circRNAs (average = 7.75%) originated from intergenic sequences (Figure 1b). Figure 1c mainly shows the distribution of circRNAs on chromosomes. It can be seen that these detected circRNAs in the current study originated from all chromosomes of cows, while the largest number of circRNAs came from chromosome 1. In addition, the number of exons contained in a circRNA is examined. From Figure 1d, it can be observed that although the majority of them consisted of multiple exons, more than 20% of circRNAs were composed of a single exon.

### 2.3. Analysis of Differential Expression Patterns of circRNAs

Using SRPBM as the metric, the overall expression of circRNAs in each cow is presented in Figure 2a. With the log_2_ (fold change) ≥ 1 and *p* < 0.05 as a threshold to screen the DE circRNAs, 140 circRNAs were differentially expressed in Sum1 and Spr1 groups (comparison1, 71 of them were up-regulated and 69 were down-regulated, Figure 2b), while 205 circRNAs were differentially expressed in Sum1 and Spr2 groups (comparison2, 83 of them were up-regulated and 122 were down-regulated, Figure 2c). A statistical summary of RNA sequencing data for each comparison is shown in Appendix A. Furthermore, 34 circRNAs were differentially expressed in both comparisons (Figure 2d, Appendix A). More circRNAs showed differential expression between Sum1 and Spr1 groups, compared to between Sum1 and Spr2 groups. A hierarchical clustering heatmap of the DE circRNAs showed that the expression patterns of the circRNAs were clearly differentiated and aggregated between Sum1 and Spr1 groups, as well as between Sum1 and Spr2 groups (Figure 2e,f). These results suggest that heat stress greatly modulated the expression of circRNAs in cows’ blood. Moreover, as for the two groups with the same cows in different lactation stage in comparison, their DE circRNAs were due to different heat stress and lactation stage, while regarding the two groups with different cows in the same lactation stage in comparison2, the DE circRNAs were mainly due to different heat stress and individuals. Accordingly, 34 overlapped DE circRNAs in 2 comparisons should be more highly correlated with heat stress, since the circRNAs’ abundance was affected by either lactation stage or individual genetic background.

### 2.4. GO and KEGG Analyses of Source Genes of DE circRNAs

In order to investigate the potential biological functions of source genes of DE circRNA, gene ontology (GO) and Kyoto Encyclopedia of Genes and Genomes (KEGG) pathway enrichment analyses were carried out with source genes of 140 and 205 DE circRNAs in the first and second comparisons, respectively. GO categories were assigned to the source genes of circRNAs, and the genes were classified into three GO categories: cellular component, biological process, and molecular function (Figure 3a and Figure 4a). Summary statistics for GO and KEGG entries for each comparison are shown in Appendix A. The current study presents only the top 50 GO terms with significance (*p* < 0.05) and the top 20 KEGG pathways. Based on the enrichment results of parental genes from DE circRNAs, RNA polymerase ΙΙ regulatory region sequence-specific DNA binding, protein import into nucleus, autophagy, and ion transport GO terms were detected in Sum1 and Spr1 groups (Figure 3a). Meanwhile, the pathways significantly changed by the DE circRNAs in Sum1 vs. Spr1 were the sphingolipid signaling pathway, cAMP signaling pathway, and PI3K-Akt signaling pathway (Figure 3b). Similar to the previous analysis, the GO and KEGG enrichment for Sum1 vs. Spr2 groups is shown in Figure 4. From Figure 4a, we can find that double-strand break repair via homologous recombination, innate immune response, and protein autophosphorylation items are most enriched. For the KEGG pathway, they are most significantly enriched in the autophagy–animal pathway, transcriptional misregulation in cancer, and the thyroid hormone signaling pathway (Figure 4b).

Meanwhile, the significantly changed pathways related to heat stress and their enriched DE circRNAs in two comparisons were visualized by using the OmicStudio tools. As shown in Figure 5, there were 15 significantly changed pathways in Sum1 vs. Spr1 groups and 9 in Sum1 vs. Spr2 groups (*p* < 0.05). These pathways were enriched for 39 parental genes. Among 15 pathways in Sum1 vs. Spr1 groups, the sphingolipid signaling pathway showed the most significance (*p* = 0.001) and contained five enriched DE circRNAs: circRNA12069 (*PTEN*), circRNA12923 (*ROCK1*), circRNA2560 (*NFKB1*), circRNA3691 (*PIK3CB*), and circRNA5055 (*PLD1*). Furthermore, circRNA2560 (*NFKB1*) and circRNA3691 (*PIK3CB*) were enriched in the longevity-regulating pathway, cAMP signaling pathway, and HIF-1 signaling pathway, which played important roles in energy metabolic regulation in cattle under heat stress. Meanwhile, among 9 pathways from Sum1 vs. Spr2 groups, the autophagy–animal pathway was the most significantly enriched functional pathway (*p* = 0.009), which included 6 DE circRNAs: circRNA1153 (*AMBRA1*), circRNA11953 (*MTMR3*), circRNA13493 (*RHEB*), circRNA5260 (*SH3GLB1*), circRNA6642 (*AKT3*), and ciRNA1282 (*HIF1A*).

### 2.5. Analyses of the Overlapped DE circRNAs in Two Comparisons

There are 34 circRNAs overlapped in 2 comparisons. The GO and KEGG annotations for these two differentially expressed circRNAs are displayed in Table 2 and Table 3. It was found that most of these differential circRNAs were enriched in pathways related to energy metabolism and RNA processing. In the biological process category, two important GO subcategories, guanyl-nucleotide exchange factor activity and cellular response to transforming growth factor beta stimulus, were annotated. In the molecular function category, the small GTPase-mediated signal transduction and repressing transcription factor binding subcategories were also annotated. The biological interpretations of the circRNA source genes were further analyzed using the KEGG pathway database, and three KEGG pathways were enriched: choline metabolism in cancer, endocytosis, and the longevity-regulating pathway. CircRNA27183 (*DOCK7*), circRNA13493 (*RHEB*), CiRNA3301 (*WWOX*), and circRNA4205 (*NR3C1*) were enriched in multiple GO and KEGG entries and may exhibit functions in response to heat stress of cattle.

### 2.6. Construction of the Protein–Protein Interaction Network and Determination of Hub Source Genes of circRNA

Using the protein interaction network, the protein interaction relationships between different genes can be determined. With the cytoNCA plug-in of Cytoscape 3.9.1 online software, we determined which source gene of circRNA was in the hub position in the regulatory network. PPI networks were constructed on the basis of STRING database. A total of 39 KEGG enriched source genes of circRNA in the network (Figure 5) and 34 overlapped source genes of circRNA (Table 3) were selected to map the PPI network using String database (http://string-db.org (accessed on 20 March 2023)) (Figure 6a). In total, 57 nodes and 47 edges were displayed in the PPI network. Finally, 26 hub source genes of circRNA were found by cytoNCA (Cytoscape plug-in) (Figure 6b). The top 10 source genes of circRNA are circRNA12069 (*PTEN*), ciRNA1282 (*HIF1A*), circRNA14021 (*PTPRC*), circRNA13493 (*RHEB*), circRNA22544 (*EZH2*), ciRNA292 (*RUNX2*), circRNA2467 (*NR3C1*), circRNA4205 (*NR3C1*), circRNA3691 (*PIK3CB*), circRNA12923 (*ROCK1*), and circRNA5055 (*PLD1*). Detailed information regarding these 10 hub genes is listed in Table 4.

### 2.7. Interaction of circRNA and miRNA

A myriad of studies have shown that circRNAs can regulate miRNA as competitive endogenous RNAs. To further understand the function of circRNAs, we used both TargetScan and miRanda to predict the interactions between circRNAs and miRNAs. Then, the interaction network data files were generated and imported into OmicStudio tools. The attributes of the target circRNAs were visualized in the network. Based on all 345 differentially expressed circRNAs, 1917 miRNAs were estimated to have a tendency to interact with circRNAs (miRanda_score ≥ 145; miranda_Energy ≤ −20 Kcal/mol, Appendix A). According to the above differential expression and GO/KEGG/PPI enrichment analyses, we identified 10 genes of different sources related to heat stress in cattle, corresponding to 11 circRNAs (Table 4). Figure 7 shows the miRNA–circRNA network diagram for all 11 circRNAs, in which circRNA292 interacted with 38 miRNAs, circRNA1282 interacted with 19 miRNAs, and circRNA2467 and 4205 interacted with 8 miRNAs. These results suggest that the identified circRNAs may be involved in heat-stress-related signaling pathways through sponging multiple miRNAs.

### 2.8. Verification of circRNA by RT-qPCR

In order to confirm the RNA-seq data, six circRNAs were randomly chosen for validation by PCR amplification with divergent primer or with one primer spanning the spliced junction. The gel electrophoresis results reveal that each circRNA had a single band at the expected length location (Figure 8a). With the intention of verifying that the observed bands were truly the corresponding circRNAs, the cDNA sequencing results were obtained. Because of the specificity of primers and the circular character of circRNAs, the back-splicing junction can be observed in circRNA products (Figure 8b). The sequencing results are in agreement with the expected RNA sequence, and the splice sites were spotted. Finally, we successfully confirmed the expressions of these six circRNAs by RT-qPCR. Results were compared with the high-throughput RNA-seq results, which showed that the expression of the six circRNAs was consistent with the trends obtained from RNA-seq data (Figure 8c).

## 3. Discussion

High ambient temperature has a profound effect on animal welfare. With escalating global temperature, compounded by the increased intensification of production, heat stress significantly impacts the dairy industry. Cows are homoeothermic animals, and their thermal comfort ranges from 5 °C to 25 °C [22]. Cows can maintain physiological stability through their own regulatory mechanisms, but when the external environmental temperature is higher than this range, cows will suffer from terrible heat stress [23,24]. Therefore, determining whether a cow is under heat stress and its response level has become a research focus. Various indicators have been established to assess the status of cows experiencing heat stress, such as the THI. Usually, THI = 72 is used as a threshold for judging the heat stress status of dairy cows. When THI is higher than 72, cows will encounter heat stress [25]. In this study, experimental cows were undergoing heat stress (Summer: THI = 81.08 ± 4.57), triggering a series of thermoregulation responses. Accompanied by reduced feed intake, increased water intake, and respiratory rate, cows exhibit a range of physiological activities [26]. The RT and RR, two important indicators of heat stress [27], were found to significantly increase (*p* < 0.05) in Sum1 cows, suggesting the involvement of physiological thermoregulation. We also found that heat stress had a significant effect on milk yield. It has been shown that cow milk production in the heat-stressed group decreased by about 10–20% compared to the group under appropriate temperature conditions [28,29], which is consistent with our results. These results reveal that Holstein cows in Beijing were undergoing severe temperature stress in July. Furthermore, heat stress had damaging effects on milk production traits and physiological condition of Holstein cows in the Sum1 group.

CircRNAs are a new class of endogenous non-coding RNAs that are covalently bound in a closed-loop structure and are more stable than linear RNAs [30]. Despite being discovered nearly 40 years ago, the significance of circular RNA has only recently been recognized, and circRNAs are now known to play a key role in gene regulation [31]. Due to their ability to bind to miRNA molecules, circRNAs can reduce the repressive effect of miRNAs on their target genes in the circRNA–miRNA–mRNA regulatory network [32]. Moreover, many reports have confirmed the role of circRNAs both in development and disease by analyzing their expression profiles [33,34]. However, little is known about the role of circular RNAs in cattle undergoing heat stress in peripheral blood. The present work investigated (i) changes in circRNAs that occurred in cattle under heat stress and (ii) key pathways and hub circRNAs involved in the cattle in response to heat stress.

To explore the regulatory effect of circRNAs on dairy cows under heat stress, we profiled the total expression of circRNAs in Holstein cattle with RNA-seq and extracted the characteristics of circRNAs in response to heat stress. According to GO and KEGG pathway analysis results, critical pathways related to the metabolic processes and autophagy in dairy cows under heat stress were identified. Although reduced dry matter intake due to heat stress has long been recognized to be the main cause of reduced milk production, a growing body of research is revealing that significant changes in the metabolism processes of cows in hot environmental conditions may be the key to milk production and milk composition [17,35]. In the present study, we found the key signaling pathways enriched in KEGG metabolic pathways in peripheral blood between the Spr and Sum groups, such as choline metabolism, the PI3K/AKT signaling pathway, the sphingolipid signaling pathway [36], and the cAMP signaling pathway [37], all of which have been previously reported to involved in heat stress response in other studies. Yang et al. [38] reported that treatment of mammary epithelial cells with choline effectively inhibited heat-stress-induced oxidative stress and apoptosis. In our results, overlapped DE in cows’ circRNAs under heat stress was most enriched in choline metabolism, which is related to alleviating heat-stress-induced damages by reducing oxidative stress [39]. Furthermore, PI3K/AKT [40], sphingolipid [36] and cAMP [37] have been proved to be important signaling molecules participating in adaptation to heat stress by affecting multi-level regulatory networks. From our results, the RT and RR significantly increased but the milk yield decreased with the THI in cows (Table 2), which revealed that Holstein cows in Beijing were undergoing severe temperature stress in July. Thus, cows tried their best to mitigate the damage caused by heat stress through a variety of metabolic processes.

Furthermore, heat stress can alter oxidative homeostasis and then cause cellular autophagy and apoptosis. This is consistent with the enriched KEGG pathways presented here: the HIF-1 signaling pathway, the longevity-regulating pathway, autophagy, and the mitophagy pathway. Heat stress was suggested to be responsible for inducing oxidative stress during summer in livestock animals. Later on, oxidative stress will result in the production of reactive oxygen species (ROS), alteration of ATP product, DNA damage, and inhibition of protein synthesis [41]. As the consumption of oxygen is elevated under thermal stress, limitations of oxygen may lead to a transient hypoxia. Additionally, the oxidative stress may activate the HIF-1 pathway [42]. Moreover, the enriched autophagy and mitophagy pathway are also highly related to oxidative stress triggered by heat stress. Baechler et al. [43] found that autophagy and mitophagy are two essential mechanisms regulating mitochondrial oxidative stress and mitochondria-associated cell death events.

Key DE circRNAs were identified as related to the metabolic processes and autophagy in cattle under heat stress. Among them, the expression of ciRNA1282 (HIF1A) was significantly changed in Sum1 compared with Spr2, and it binds to multiple miRNAs. Chen et al. [44] found that Hsp90 relieves heat-stress-induced damage by involvement of autophagic HIF-1α signaling, which suggested that HIF1A might alleviate the damage caused by heat stress in animals. Moreover, ciRNA1282 (HIF1A) might bind to the bta-let-7 family, in which Bta-let-7c has been shown to promote adipocyte proliferation and inhibit adipocyte differentiation by acting as a sponge for circFUT10 [45]. Furthermore, circRNA13493 (RHEB) and circRNA4205 (NR3C1) were differentially and consistently expressed in two groups. RHEB is a small H-Ras-like GTPase that is a direct and essential activator of the lysosomal surface rapamycin complex 1 (mTORC1) machinery, regulates cell growth, and activates autophagy [46,47]. Heat stress may activate the modulatory effects of RHEB on the mTOR pathway, thereby registering proliferation and autophagy in dairy mammary cells. As for NR3C1 (glucocorticoid receptor), it was found to reduce the promoter methylation under heat stress in dairy cattle [48]. Another interesting finding is that the expression of circRNA22544 (EZH2) significantly differed between Sum1 and Spr1 [16]. Wang et al. [16] found that heat-stress-related circEZH2 affected the proliferation, apoptosis, and lipid metabolism of mammary gland epithelial cells, which is consistent with our results.

Usually, in the investigation of heat stress of cattle, with the increase in experimental samples’ quantity, the analysis is statistically more robust and reliable. However, sampling and sequencing require a lot of manpower and expenses. In practice, generally more than three samples per group are selected for analysis [49,50]. In this manner, we randomly selected 5 animals per group of 15 animals for RNA-seq. Moreover, the comparison design can be constructed either by using cows with the same lactation stage in different temperatures, such as Spr2 and Sum1 in this study, or by using the same samples assessed in two seasons, such as Spr1 and Sum1 in this case (apparently, the lactation stage of samples is different). Between Sum1 and Spr1 groups, pathways closely related to energy metabolic and oxidative stress were found, such as the sphingolipid signaling pathway, cAMP signaling pathway, PI3K-Akt signaling pathway and HIF-1 signaling pathway. Liu et al. [21] selected five Holstein cows (female, healthy) and extracted blood samples at the same moment by coccygeal venipuncture in the summer (August) and winter (December) for RNA-seq. Their functional analyses showed that the MAPK signaling pathway, cellular senescence cGMP-PKG signaling pathway, and thermogenesis are related to oxidative damage and cell apoptosis. This result suggested that heat stress might have different impacts on cows in different stages of lactation. This might be due to the fact that heat stress can lead to an aggravation of lipid peroxidation and a decrease in total antioxidant capacity in dairy cows at different lactation stages. Moreover, heat stress negatively impacts feed intake, milk yield, and negative energy balance in early lactation [51]. However, in Sum1 and Spr2 groups, the largest number of DE circRNAs were enriched in heat-stress-related signaling pathways such as autophagy. Luo et al. [2] selected 8 primiparous (4 cows in April, 4 cows in July) cows with DIM ranging from 135 to 144 d (mid-lactation and pregnant) for RNA-seq. In their study, some genes associated with autophagy and apoptosis were identified. It was pointed out by Zhang et al. [52] that exposure of germ cells to hyperthermia resulted in several specific features of the autophagic process. Therefore, heat stress might mainly affect the expression of circRNA through autophagy in dairy cows at the same lactation stage and thus induce oxidative injury in paired samples.

From the above analysis, it can be seen that different comparison designs will lead to different results. It is important for researchers to choose an appropriate comparison design for heat stress study. When dairy cows are in lactation stage, they are more likely to be affected by heat stress. Consequently, their energy consumption is higher and their metabolic activity is more active [53]. When we are more concerned about the effects of heat stress on cows’ lactation performance, the impact on cow lactation cannot be ignored. In this case, comparison1 is probably a better choice. However, supposing that we are more concerned about the effects of heat stress on the cow as an organism, we should weigh the impact of different lactation stages on cows. Moreover, the breed of dairy cattle is also relevant to the choice of experimental samples. Holstein cows display fewer individual differences than local breeds, and thus, they are more appropriate for the design of comparison2. Our results provide the experimental design suggestions and basis to study dairy cows’ responses to heat stress.

## 4. Materials and Methods

### 4.1. Animals and Sampling

The experiment was performed in agreement with the Committee on Ethics of Animal Experimentation at the Beijing Jiaotong University, Beijing, China (Code ID: SS-QX-2014-06; 26 June 2014).

A total of 30 healthy primiparous Holstein lactating cows with similar age and body weight from a commercial farm (Beijing, China) were selected as experimental animals. All cows were raised in a loose pen with an exercise yard outside the house, free access to water, and a total mixed ration (TMR) throughout the day. What is more, the diet’s nutrient composition remained consistent throughout the test period, and its nutritional level was in accordance with the standard for dairy cattle feeding (NY/T 34-2004). Appendix A shows the composition and nutritional level of the feed. The temperature and humidity were automatically recorded every half hour. The temperature–humidity index (THI) was calculated according to the equation THI = 0.8AT + RH × (AT − 14.1) + 46.4 (AT and RH are temperature and relative humidity) [54]. Out of 30 cows, the Spr1 group (15 cows) and the Spr2 group (15 cows) were sampled in April 2017 (the thermoneutral period, THI = 55.43), and the Sum1 group (same 15 cows as in Spr1) was sampled in July 2017 (heat stress period, THI > 78). Among them, Spr1 (DIM = 61.6 ± 10.62 d) and Sum1 (DIM = 137.60 ± 10.62 d) were the same individuals, while Spr2 (DIM = 137.53 ± 7.63 d) and Sum1 (DIM = 137.60 ± 10.62 d) were in the same lactation stage.

### 4.2. Measurement of Phenotypic Data

During the experimental periods, daily milk yield (MY) for all 45 cows were collected and the average milk yield for 7 days (7AMY) was calculated. Rectal temperature (RT) was measured 3 times a day using a digital thermometer with a precision of 0.1 °C (MC-347, OMRON, Tokyo, Japan) by leaving the animal thermometer in the rectum for 10 s in the morning (0700–0900 h), afternoon (1400–1600 h), and evening (2100–2300 h) for 3 continuous days. At the same time, the breathing rate was measured by the flanking movement of the cow’s body for 30 s by visual inspection and multiplied by 2 to determine breaths per minute (breaths/min).

### 4.3. Blood Sample Collection and Total RNA Extraction

Blood samples (10 mL) for each cow were collected into ETDA2K anticoagulant tubes via tail vein puncture and centrifuged at 1400× *g* for 15 min at 4 °C. The middle leukocyte was extracted and stored at −80 °C for further RNA extraction. Following the protocol of the manufacturer, Trizol reagent (Invitrogen, Carlsbad, CA, USA) was employed for total RNA isolation and purification. RNA integrity was assessed using 1% agarose gel electrophoresis. The quality and quantity of RNA were assessed through NanoDrop ND-1000 (NanoDrop, Wilmington, DE, USA) and Agilent 2100 Bioanalyzer (Agilent, Santa Clara, CA, USA). Only when RNA integrity number was >7.0 and RNA electrophoresis results (28S/18S ≥ 1.0), the RNAs could be used for further analysis.

### 4.4. Library Construction and RNA Sequencing

In each group, five cows were selected randomly from each group for RNA-seq. Sequencing libraries were generated following the manufacturer’s recommendation, and index codes were added to assign sequences to each sample. Afterwards, in accordance with the instructions of RiboZeroTM rRNA Removal Kit (Illumina, San Diego, CA, USA), ribosomal RNA (rRNA) was depleted. After the depletion of rRNA, the remaining RNAs were fragmented into small pieces and later reverse-transcribed to construct the cDNA library. Implementations include synthesizing U-labeled second-stranded DNAs with E. coli DNA polymerase I, RNase, and dUTP; adding A-base; ligating the adapter; and some other operations. Finally, llumina Hiseq 4000 (LC Bio, Hangzhou, Zhejiang, China) platform was used to perform the pair-end sequencing according to the vendor’s recommended protocol.

### 4.5. Circular RNA Identification and Differential Expression Analysis

Firstly, raw data obtained were filtered by Cutadapt [55], which could remove the reads that contained adaptor contamination, low-quality bases (Q ≥ 10), and undetermined bases. Then, in order to ensure the accuracy of subsequent analysis, sequence quality was evaluated by using FastQC (http://www.bioinformatics.babraham.ac.uk/projects/fastqc/ (accessed on 8 February 2022)). After that, Bowtie2 [56] and Hisat2 [57] were applied to map reads to the Bos taurus reference genome. Later on, the remaining reads (58 unmapped reads) were extracted and mapped to genome using tophat-fusion [58]. With the aim of predicting circRNA more credibly, at the beginning, CIRCExplorer2 [59,60] and CIRI [61] were used to de novo assemble the mapped reads to circular RNAs. Then, tophat-fusion was used again to identify the back-splicing reads. Only the circRNA with at least one back-splicing read can be confirmed. To estimate the relative expression level of a circRNA, its quantity is denoted as spliced reads per billion mapped reads (SRPBM). The DE circRNAs were calculated between different treatment groups with |log_2_(fold change)| ≥ 1, *p* < 0.05 by R package edgeR [62].

### 4.6. GO and Pathway Analyses of Source Genes of DE circRNAs

The gene ontology (GO) and Kyoto Encyclopedia of Genes and Genomes (KEGG) pathway enrichment analyses were conducted to explore the potential function of circRNAs. The gene corresponding to the parental mRNA of circRNA was identified with terms in the GO and KEGG database. In this manner, the function of gene was classified by GO pattern and KEGG pathway, and then all differentially expressed circRNAs could be mapped into terms of GO and KEGG. After comparing with the genome background using Fisher’s exact test, the significantly enriched (*p* < 0.05) GO and KEGG terms are kept for further analysis. OmicStudio tools were used to visualize the significantly changed heat-stress-related pathways and their enriched DE circRNAs in two comparisons.

### 4.7. Protein–Protein Interaction (PPI) Network Construction of Hub Source Genes of DE circRNAs

The String online database (https://cn.string-db.org/ (accessed on 20 March 2023)) provides some reliable information on PPI [63]. In this study, minimum required interaction score = 0.400 was considered. After extracting information on PPI, CytoNCA in Cytoscape was used for network visualization. Cytoscape is an open source software platform for visualizing complex networks and integrating these with any type of attribute data [64]. CytoNCA is a Cytoscape plugin for centrality analysis and evaluation of protein interaction networks [65] and is used to identify the hub source genes of DE circRNAs obtained from the PPI network.

### 4.8. Construction of circRNA–miRNA Network

Previous study hypothesized that by acting as miRNA sponge, circRNAs can bind miRNAs through an miRNA response element (MRE) and negatively regulate their activity [31]. The co-expressed circRNA and miRNA have potential as biomarkers and therapeutic targets. To uncover this co-expression network, we used miRanda [66] and Targetscan [67] to determine the interaction between circRNA and miRNA.

### 4.9. Quantitative Real-Time PCR Validation

Six circRNAs (four up-regulated and two down-regulated) were selected randomly to perform qRT-PCR to confirm the results of RNA-seq. In brief, 500 ng total RNA corresponding to the 15 samples was reverse-transcribed to cDNA. The expression levels of the 6 circRNAs were investigated by qPCR using Eastep R qPCR Master Mix (Promega, Shanghai, China), and the final result was observed by harnessing ABI Prism 7900HT sequence detection system (Thermo Fisher, Waltham, MA, USA). The primers were designed for RT-qPCR using circPrimer and Primer3. The circRNA and their corresponding primer sequences are provided in Appendix A. Biological replicates were considered for all the quantitative PCR reactions. Afterward, using the designed primers and cDNA template, a polymerase chain reaction was conducted. The PCR conditions were 94 °C denaturation for 5 min, 40 cycles at 94 °C for 10 s, 54 to 60 °C for 15 s, and 72 °C for 30 s. The relative expression levels of circRNA were calculated using the 2^−∆∆Ct^ method. The GAPDH was used as an internal control. Finally, agarose gel electrophoresis and DNA sequencing by Sanger sequencing were employed to confirm the qPCR products.

### 4.10. Statistical Analysis

The quantitative data were analyzed using SPSS software (ver. 24.0, SPSS Inc., Chicago, IL, USA), and graphs were generated using GraphPad Prism 7.0 software (ver. 5.03, GraphPad Software, San Diego, CA, USA). Data were expressed as mean ± standard error of the mean (s.e.m). Significant differences were determined by Student’s *t*-test or one-way ANOVA. *p* values < 0.05 were considered statistically significant differences unless otherwise stated.

## 5. Conclusions

In this study, two comparisons consisting of three experimental groups in spring and summer were designed to identify the effect of regulation of circRNA on heat stress. It can be seen that heat stress drastically reduced the overall milk yield and damaged the physiological condition of Holstein cows in Beijing in July. There were significant changes in circRNA expression profiles in peripheral blood samples from cattle under heat stress. Key pathways related to metabolic processes and autophagy such as the HIF-1 signaling pathway and longevity-regulating pathway were identified as participating in the cattle heat stress response. In addition, heat-stress-responsive circRNAs and their targeting miRNAs or corresponding genes were found. This valuable information will enrich our knowledge about involvement of key circRNAs and their expression patterns in cows’ heat stress processes. Moreover, our study can be further utilized in the identification, characterization, and designation of breeding strategies to develop both high-yielding and thermotolerant Holstein dairy cows.

## Figures and Tables

**Figure 1 ijms-24-10150-f001:**
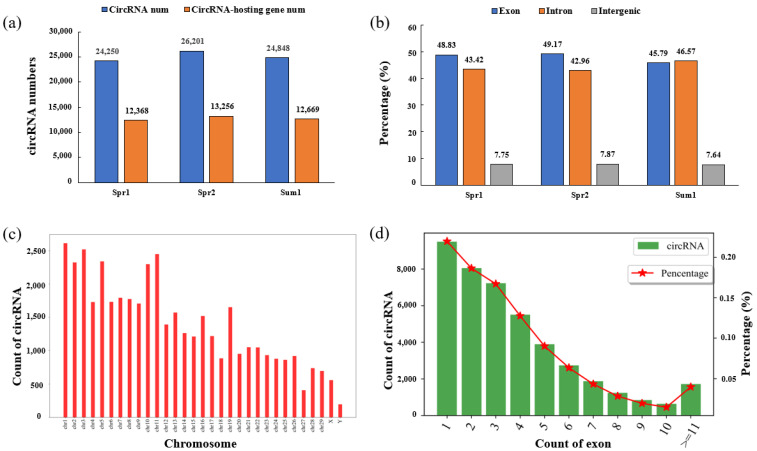
Genomic characteristics of circRNA identified in Holstein cattle. (**a**) CircRNA numbers predicted in each group. (**b**) CircRNAs from exons, introns, or intergenic regions. (**c**) Distribution of circRNAs on chromosomes. (**d**) Number of exons contained within exonic circRNAs.

**Figure 2 ijms-24-10150-f002:**
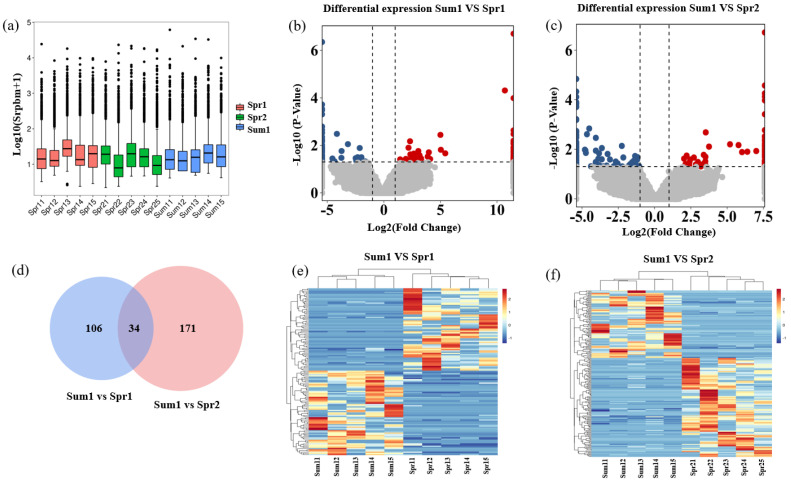
DE circRNAs in response to heat stress in Holstein cattle. (**a**) Distribution of circRNA expression in 15 samples. (**b**,**c**) The volcano plot of DE circRNAs detected in two comparisons. Red dots represent the up-regulated circRNAs, blue dots represent the down-regulated circRNAs, and gray dots represent the unchanged circRNAs. (**d**) Venn diagram of DE circRNAs in comparison1 (Sum1 vs. Spr1) and comparison2 (Sum1 vs. Spr2) data sets. (**e**,**f**) The heatmaps of DE circRNAs in two comparisons.

**Figure 3 ijms-24-10150-f003:**
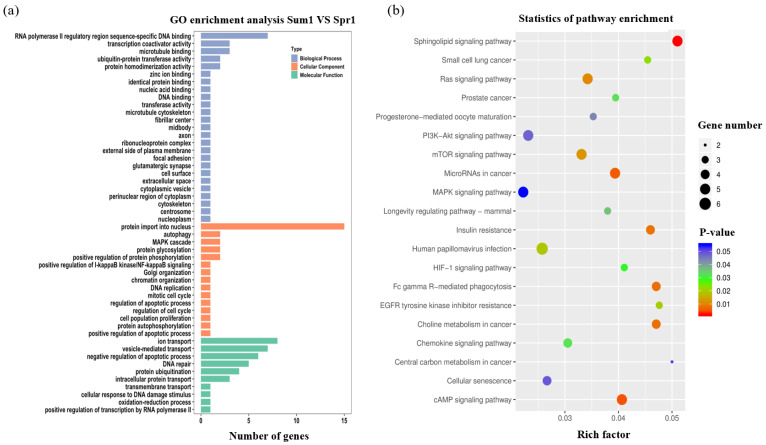
Annotations and enrichment analysis of the DE circRNAs in Sum1 and Spr1 groups. (**a**) GO annotations of DE circRNA host genes. Diagram shows the significantly enriched GO terms (*p* < 0.05). The abscissa represents the terms in GO categories of biological process (BP), cellular component (CC), and molecular function (MF). (**b**) KEGG enrichment analysis of DE circRNAs. The abscissa indicates the rich factor, and the ordinate indicates the name of the enriched pathway. The size of the dot represents the number of enriched genes in the pathway, and the color corresponds to the *p* value as indicated in the figure legend.

**Figure 4 ijms-24-10150-f004:**
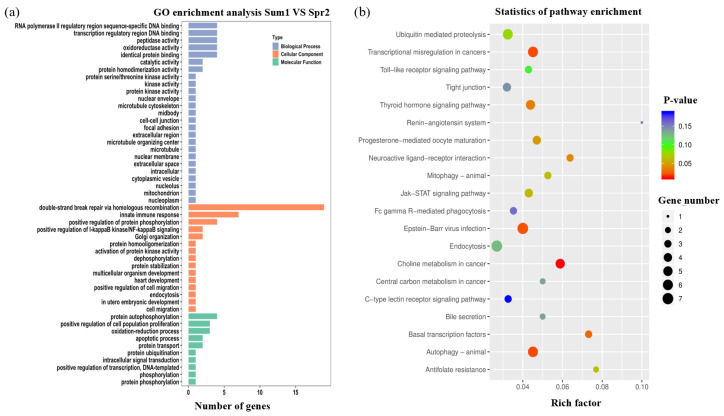
Annotations and enrichment analysis of the DE circRNAs in Sum1 vs. Spr2 groups. (**a**) GO annotations of DE circRNA host genes. Diagram shows the significantly enriched GO terms (*p* < 0.05). The abscissa represents the terms in GO categories of biological process (BP), cellular component (CC), and molecular function (MF). (**b**) KEGG enrichment analysis of DE circRNAs. The abscissa indicates the rich factor, and the ordinate indicates the name of the enriched pathway. The size of the dot represents the number of enriched genes in the pathway, and the color corresponds to the *p* value as indicated in the figure legend.

**Figure 5 ijms-24-10150-f005:**
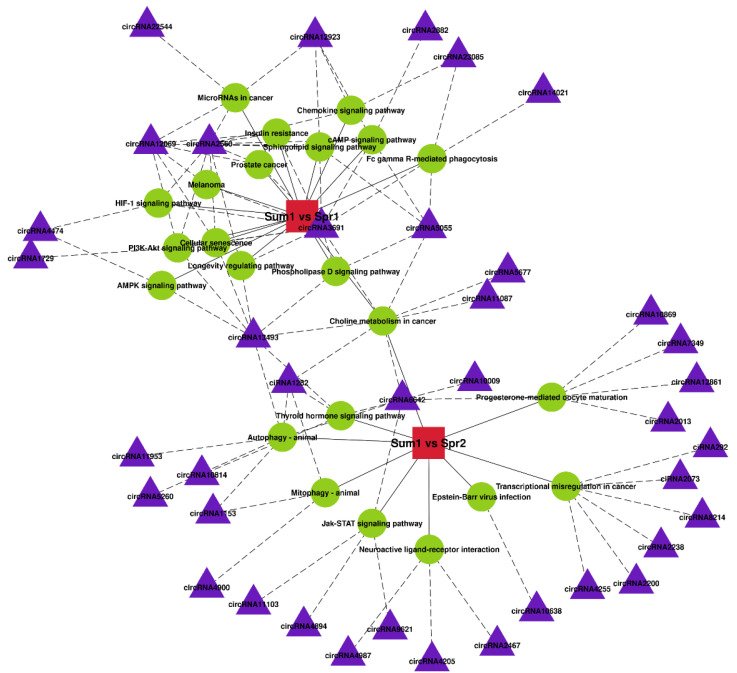
Relationship between significant pathways and enriched DE circRNAs in two comparisons. The red square nodes represent comparison1 (Sum1 vs. Spr1) and comparison2 (Sum1 vs. Spr2), the green circular nodes represent pathways, the purple triangle nodes represent DE circRNAs, the solid lines represent significantly enriched pathways in comparisons, and the dotted lines represent DE circRNAs involved in enriched pathways.

**Figure 6 ijms-24-10150-f006:**
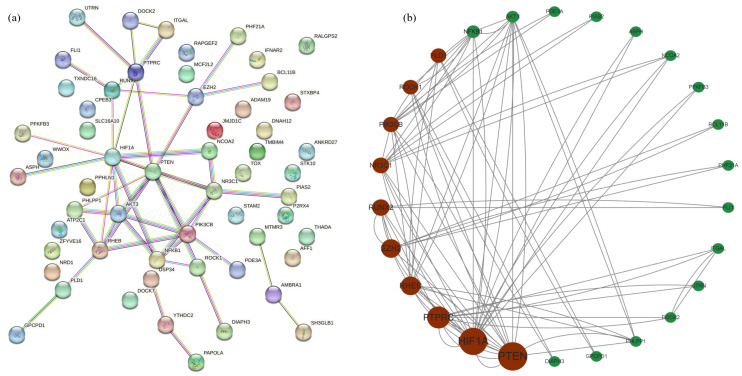
Identification of hub source genes of circRNA from the PPI network. (**a**) PPI network of 74 source genes of circRNA. (**b**) PPI network of 26 source hub genes of circRNA that were extracted from the PPI network. The larger the circle, the higher the Betweenness Centrality (BC); conversely, the smaller the circle, the lower the BC. The red circles note the top 10 hub source genes of circRNA.

**Figure 7 ijms-24-10150-f007:**
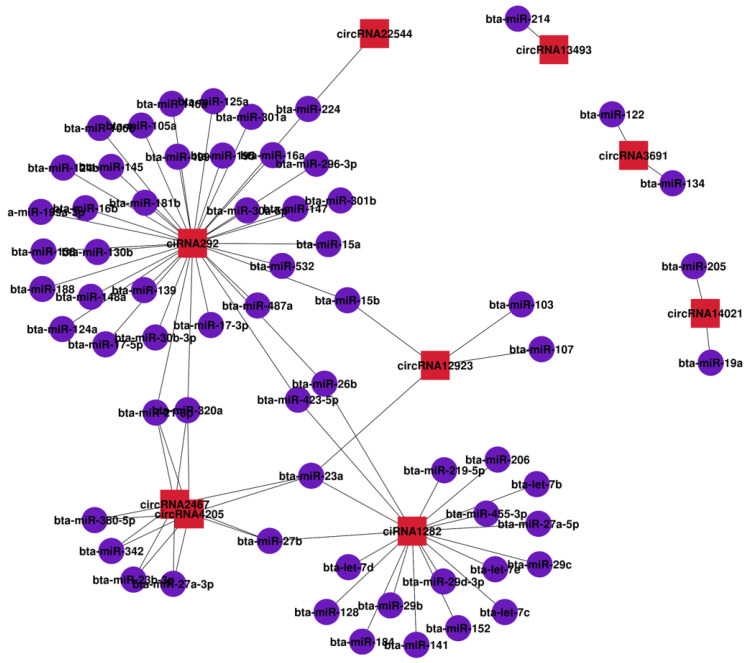
Interaction between circRNAs and miRNAs. The red squares represent circRNA, the purple circles represent miRNA, and the lines represent the relationships between circRNAs and miRNAs.

**Figure 8 ijms-24-10150-f008:**
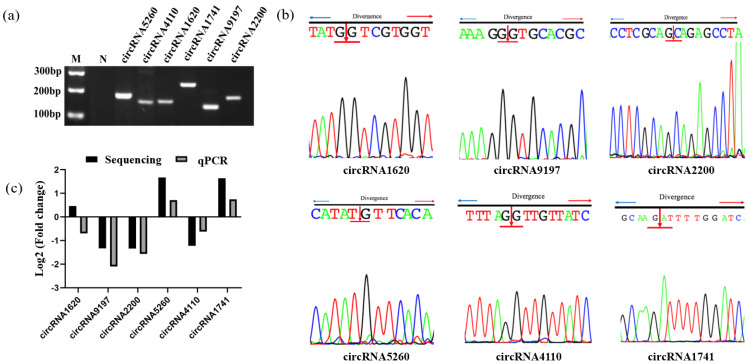
Validation of DE circRNAs. (**a**) The gel electrophoresis results of circRNAs. (**b**) Sanger sequencing confirmed the back-splice junction sequence of the indicated circRNAs. (**c**) Comparison of results from RNA-seq and qRT-PCR. Six DE circRNAs were randomly selected, and their expression level was validated by qRT-PCR.

**Table 1 ijms-24-10150-t001:** The performances and physiological indicators of Holstein cattle in Spr1, Spr2, and Sum1 groups.

Variable	Spr		Sum	*p*-Value ^3^
	Spr1 (MEAN ± SD) ^1^	Spr2 (MEAN ± SD) ^1^	Sum1 (MEAN ± SD) ^1^	
Individual	15	15	15 ^2^	NA
DIM (d)	61.6 ± 10.62	137.53 ± 7.63	137.60 ± 10.62	NA
7AMY (Kg/day)	40.91 ± 4.40 ^ab^	42.93 ± 3.25 ^a^	38.99 ± 3.64 ^b^	<0.05
MY (Kg)	43.95 ± 3.89 ^a^	40.88 ± 2.05 ^b^	37.51 ± 5.36 ^c^	<0.05
RT (°C)	38.63 ± 0.18 ^b^	38.58 ± 0.13 ^b^	38.96 ± 0.24 ^a^	<0.05
RR (breaths/min)	38.80 ± 3.63 ^b^	37.60 ± 5.14 ^b^	88.4 ± 12.39 ^a^	<0.05
THI	54.96 ± 3.27		81.08 ± 4.57	<0.05

Note: DIM, days in milk; 7AMY, average milk yield across seven days; MY, milk yield on the blood sampling day; RT, rectal temperature; RR, respiration rate; THI, temperature–humidity index; ^1^ Mean ± standard deviation; ^2^ they are the same individuals as in Spr1; ^a,b,c^ Different letters in the same row indicate significant differences between the two groups (*p* < 0.05); ^3^ *p* < 0.05 indicates a significant difference among groups; NA: not applicable.

**Table 2 ijms-24-10150-t002:** The GO term of overlapped circRNAs in two comparisons.

GO Term	CircRNA (Source Gene)	*p*-Value ^1^
Guanyl-nucleotide exchange factor activity	CircRNA27860 (*ANKRD27*); circRNA27332 (*RALGPS2*); circRNA3628 (*MCF2L2*); **circRNA27183 (*DOCK7*)**	0.001
Cellular response to transforming growth factor beta stimulus	**CiRNA3301 (*WWOX*)**; **circRNA4205 (*NR3C1*)**	0.045
Small GTPase-mediated signal transduction	CircRNA27332 (*RALGPS2*); **circRNA13493 (*RHEB*)**; **circRNA27183 (*DOCK7*)**	0.001
Repressing transcription factor binding	CiRNA292 (*RUNX2*); **circRNA4205 (*NR3C1*)**	0.001
Cytosol	circRNA27437 (*PPHLN1*); **circRNA4205 (*NR3C1*)**; circRNA11103 (*STAM2*); **circRNA13493 (*RHEB*)**; ciRNA3301 (*WWOX*)	0.001

Note: Words in bold indicate the DE circRNAs were enriched in multiple GO and KEGG entries. ^1^ *p*-values were calculated using Student’s *t*-test.

**Table 3 ijms-24-10150-t003:** The KEGG term of overlapped circRNAs in two comparisons.

KEGG Term	CircRNA (Source Gene)	*p*-Value ^1^
Choline metabolism in cancer	**CircRNA13493 (*RHEB*)**; circRNA5677 (*GPCPD1*)	0.005
Endocytosis	CircRNA11103 (*STAM2*); circRNA27501 (*ZFYVE16*)	0.025
Longevity-regulating pathway	**CircRNA13493 (** ** *RHEB* ** **)**	0.0057

Note: Words in bold indicate the DE circRNAs were enriched in multiple GO and KEGG entries. ^1^ *p*-values were calculated using Student’s *t*-test.

**Table 4 ijms-24-10150-t004:** The top 10 hub source genes of DE circRNA in 2 comparisons.

Accession	Source Gene	Sum1 vs. Spr1		Sum1 vs. Spr2	
		Regulation	*p*-Value ^1^	Regulation	*p*-Value ^1^
CircRNA1209	*PTEN*	down	0.01	down	0.15
CiRNA1282	*HIF1A*	down	0.45	down	0.01
CircRNA14021	*PTPRC*	up	0.04	up	0.47
CircRNA13493	*RHEB*	down	0.01	down	0.04
CircRNA22544	*EZH2*	up	0.03	up	0.40
CircRNA292	*RUNX2*	down	0.02	down	0.00
CircRNA2467	*NR3C1*	down	0.44	down	0.05
CircRNA4205	*NR3C1*	down	0.01	down	0.00
CircRNA3691	*PIK3CB*	down	0.02	down	0.09
CircRNA12923	*ROCK1*	down	0.01	down	0.39
CircRNA5505	*PLD1*	down	0.01	down	0.39

Note: ^1^ *p*-values were calculated using Student’s *t*-test.

## Data Availability

All the pertinent data are presented in the manuscript and associated Appendix A. Raw sequencing data can be obtained from the corresponding author.

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
