# Peer review of "Analysis of CircRNA Expression in Peripheral Blood of Holstein Cows in Response to Heat Stress"

_ijms, 2023, doi:10.3390/ijms241210150_

Round 1
Reviewer 1 Report
I congratulate the authors on the subject and the quality of the article.
It presents an adequate introduction to the theme;
a correct explanation of the methodology (although I consider the sample somewhat reduced, in number and time of year, as well as I would like some complementary information about the type of installations);
adjusted and well explained results;
a complete and precise discussion that supports the whole experimental design and the scientific value of the work;
a precise conclusion;
Small corrections of the work are also suggested, namely in editing details (line 522, 667).
Author Response
We are very grateful for your time and efforts regarding our manuscript. Your suggestions are very important to us, both for composing the manuscript and our further research. We have paid careful attention to all comments and suggestions and revised the manuscript accordingly. All the changes are highlighted using the “Track Changes”. The detailed responses listed point by point are as follows.
Point 1: I congratulate the authors on the subject and the quality of the article. It presents an adequate introduction to the theme; a correct explanation of the methodology (although I consider the sample somewhat reduced, in number and time of year, as well as I would like some complementary information about the type of installations).
Response 1: Thank you very much for your appreciation of the work. Your valuable comments are highly appreciated. In the investigation of heat stress on cattle, with the increase of experimental samples, the analysis is statistically more robust and reliable, but, in the same time, more manpower and expenses will be required. So, we randomly selected 5 animals from each per group of 15 animals for RNA-seq. Moreover, we further summarized the performances and physiological indicators of five cows that performed RNA-seq in each per group. These results revealed that Holstein cows were undergoing severe temperature stress in Summer. This indicated that our experimental sample selection was reasonable. We also added some complementary information about the type of installations in the Materials and Methods section. All the changes are highlighted in the revised manuscript.
Table 1 The performances and physiological indicators for RNA-seq
Variable |
Spr1 (MEAN ± SD) 1 |
Spr2 (MEAN ± SD) 1 |
Sum1 (MEAN ± SD) 1 |
P-value 3 |
Individual |
5 |
5 |
5 2 |
NA |
DIM (d) |
66.40 ± 5.81 c |
139.00 ± 4.82 b |
142.40 ± 5.81 a |
NA |
7AMY (Kg/day) |
46.66 ± 4.37 a |
40.42 ± 1.35 b |
36.64 ± 1.877 c |
< 0.05 |
MY (Kg) |
45.68 ± 4.65 a |
40.30 ± 1.90 b |
34.02 ± 4.13 c |
< 0.05 |
RT (℃) |
38.78 ± 0.06 ab |
38.49 ± 0.16 b |
39.17 ± 0.55 a |
< 0.05 |
RR (breaths/min) |
37.20 ± 1.92 b |
40.80 ± 2.86 b |
93.87 ± 13.88 a |
< 0.05 |
THI |
54.96 ± 3.27 |
81.08 ± 4.57 |
< 0.05 |
Note: DIM, days in milk; 7AMY, average milk yield across seven days; MY, milk yield on the blood sampling day; RT, rectal temperature; RR, respiration rate; THI, temperature-humidity index; 1 Mean ± standard deviation; 2 they are the same individuals as Spr1; a, b, c Different letters in the same row indicate significant differences between the two groups (P < 0.05); 3 P < 0.05 indicates a significant difference among groups; NA: Not applicable.
Point 2: Small corrections of the work are also suggested, namely in editing details (line 522, 667).
Response 2: Thank you for this great observation. We have corrected the parts of the manuscript which were not edited correctly in the previous version. Thank you again.
Reviewer 2 Report
IJMS-2411011 - Analysis of CircRNA Expression in Peripheral Blood of Holstein Cows in Response to Heat Stress- Review Report
General comments
Generally, the paper is well-written and backed by recent publications. However, I have highlighted areas where the paper could be improved upon under the different topics.
Abstract: The abstract should be a brief report of the research work. Many readers hardly read further than the abstract. They view the abstract as a short form of the paper. The silent points from the paper should be presented as a summary in the abstract. A precise statement about the methods is always needed. Most important is a concise summary of the results. The conclusion should put the work in perspective and entice readers to read the entire work.
In the light of the above, the first 4 sentences (Lines 13-17) should be discarded or incorporated into the introduction.
Line 13 The phrase ‘rise of global temperature’ may be wordy; for clarity, consider changing it to “global temperature rise’.
Line 14 The word ‘negative’ is often overused. Consider using a more specific synonym to improve the clarity of the sentence. Suggestion: you may use the word ‘damaging’.
Line 15 It appears that a pronoun is missing after behind. Suggestion: Add the pronoun ‘it’ after behind.
1. Introduction
Line 41 The preposition (of) used here is incorrect. Suggestion use the preposition, ‘in’ instead of ‘of’.
Line 58 There is a pronoun problem here. Suggestion: Use ‘their’ instead of ‘its’.
Line 59-61 The sentence appears long and difficult to follow; consider breaking it into two. Suggestion: Subsequent research tried to uncover the functional role of circRNAs in various species and tissues. A handful of circRNAs have been demonstrated to have functions in post-transcriptional regulation, acting as miRNAs sponge [9].
Line 72 ‘number of’ is unnecessary; consider removing it.
Line 73 ‘that’ is unnecessary; consider removing it.
Line 75-77 The sentence is unclear and difficult to follow. There is a need to recast it. Suggestion: Also, copious changed circRNAs were discovered in cow mammary gland tissues [15, 17], hypothalamic, and pituitary tissues [18] under heat stress.
Line 86-89 A knowledgeable audience might find this sentence difficult to understand, it may be necessary to break it into two. Suggestion: Liu et al. described the 86 expression patterns of mRNA and miRNA in the blood of the same individual cows under different conditions, from heat stress to thermal equilibrium [21]. They suggested individuals' influence on cows' response to heat stress.
Line 93-94 Your sentence may be unclear or hard to follow; consider recasting it. Suggestion: Here, we examined the changes in milk yield, rectal temperature, and respiratory rate of dairy cows under heat stress.
2. Results
Line 131 The word ‘showed’ is in the wrong tense; change it to the right form. Suggestion: use ‘shows’ instead.
Line 176-177 Your sentence may be unclear or hard to follow; consider recasting it. Suggestion: The current study presented only the top 50 GO terms with significance (P < 0.05) and the top 20 KEGG pathways.
3. Discussion
Line 306-308 This sentence is unclear or hard to follow; consider recasting it. Suggestion: With escalating global temperature, compounded by the increased intensification of production, heat stress significantly impacts the dairy industry.
Line 308-309 Give a reference for this sentence.
Line 316-317 The sentence needs to be reworded to make it clearer to your readers. Suggestion: In this study, experimental cows were undergoing heat stress (Summer: THI = 81.08 ± 4.57), triggering a series of thermoregulation.
Line 322-324 Consider rephrasing the sentence to make it easier for your readers to follow. Suggestion: It has been shown that cows' milk production in the heat-stressed group decreased by about 10-20% compared to the group under appropriate temperature conditions [27, 28], which is consistent with our results.
Line 334-335 the sentence’s structure could be improved upon to make it clearer to readers. Suggestion: Moreover, many reports have confirmed the role of circRNAs both in development and disease by analyzing their expression profiles [32, 33].
Line 341-343 There is a need to recast this sentence to make it clearer. Suggestion: According to GO and KEGG pathway analysis results, critical pathways related to the metabolic processes and autophagy in dairy cows under heat stress were identified.
Line 347-351 Please give reference(s) for your readers to follow.
Line 375 The phrase ‘in comparison with’ should be changed to ‘compared with’.
Line 388-390 This sentence requires recasting to make it clearer to your readers. Suggestion: Another interesting finding is that circRNA22544 (EZH2) expression significantly differed between Sum1 and Spr1.
Line 402-403 This sentence is incomplete; consider rephrasing it with another sentence.
Line 403-405 The sentence is unclear and needs to be reworded to improve its clarity. Suggestion: Moreover, heat stress negatively impacts feed intake, milk yield, and energy negative balance in early lactation [48].
4. Materials and Methods
The authors stated in lines 434-436 that the experimental animals were fed a total mixed ratio (TMR) ad lib. It is essential at this point to expand by highlighting the nutritive composition of this ration as stated by the standard for dairy cattle feeding. This should be tabulated. This will enable your readers to follow the discussion better and for this study to be replicated elsewhere.
Line 443-446 The sentence appears informal and hard to follow. Rewrite it to improve its clarity. Suggestion: What is more, the diet's nutrient composition remained consistent throughout the test period, and its nutritional level was in accordance with the standard for dairy cattle feeding (NY/T 34-2004).
Line 490 The word ‘correspondent’ doesn’t seem to fit this context. Consider replacing it with a more appropriate word. Suggestion: ‘corresponding’.
Line 510 The phrase ‘In pursuit of uncovering the’, appears unclear; consider changing it. Suggestion: You may use ‘To uncover’ instead.
Conclusions: This section is not mandatory in the IJMS, but conclusions can be added to the manuscript. Some readers read the abstract and rush to read the conclusions only in a paper; if these are well written, they are encouraged to read the whole paper. Think about adding it.
References
All the IMJS and MDPI guidelines on references have not been followed and adhered to. Authors should go through all of them and make the necessary adjustments accordingly.
IJMS-2411011 - Analysis of CircRNA Expression in Peripheral Blood of Holstein Cows in Response to Heat Stress- Review Report
General comments
Generally, the paper is well-written and backed by recent publications. However, I have highlighted areas where the paper could be improved upon under the different topics.
Abstract: The abstract should be a brief report of the research work. Many readers hardly read further than the abstract. They view the abstract as a short form of the paper. The silent points from the paper should be presented as a summary in the abstract. A precise statement about the methods is always needed. Most important is a concise summary of the results. The conclusion should put the work in perspective and entice readers to read the entire work.
In the light of the above, the first 4 sentences (Lines 13-17) should be discarded or incorporated into the introduction.
Line 13 The phrase ‘rise of global temperature’ may be wordy; for clarity, consider changing it to “global temperature rise’.
Line 14 The word ‘negative’ is often overused. Consider using a more specific synonym to improve the clarity of the sentence. Suggestion: you may use the word ‘damaging’.
Line 15 It appears that a pronoun is missing after behind. Suggestion: Add the pronoun ‘it’ after behind.
1. Introduction
Line 41 The preposition (of) used here is incorrect. Suggestion use the preposition, ‘in’ instead of ‘of’.
Line 58 There is a pronoun problem here. Suggestion: Use ‘their’ instead of ‘its’.
Line 59-61 The sentence appears long and difficult to follow; consider breaking it into two. Suggestion: Subsequent research tried to uncover the functional role of circRNAs in various species and tissues. A handful of circRNAs have been demonstrated to have functions in post-transcriptional regulation, acting as miRNAs sponge [9].
Line 72 ‘number of’ is unnecessary; consider removing it.
Line 73 ‘that’ is unnecessary; consider removing it.
Line 75-77 The sentence is unclear and difficult to follow. There is a need to recast it. Suggestion: Also, copious changed circRNAs were discovered in cow mammary gland tissues [15, 17], hypothalamic, and pituitary tissues [18] under heat stress.
Line 86-89 A knowledgeable audience might find this sentence difficult to understand, it may be necessary to break it into two. Suggestion: Liu et al. described the 86 expression patterns of mRNA and miRNA in the blood of the same individual cows under different conditions, from heat stress to thermal equilibrium [21]. They suggested individuals' influence on cows' response to heat stress.
Line 93-94 Your sentence may be unclear or hard to follow; consider recasting it. Suggestion: Here, we examined the changes in milk yield, rectal temperature, and respiratory rate of dairy cows under heat stress.
2. Results
Line 131 The word ‘showed’ is in the wrong tense; change it to the right form. Suggestion: use ‘shows’ instead.
Line 176-177 Your sentence may be unclear or hard to follow; consider recasting it. Suggestion: The current study presented only the top 50 GO terms with significance (P < 0.05) and the top 20 KEGG pathways.
3. Discussion
Line 306-308 This sentence is unclear or hard to follow; consider recasting it. Suggestion: With escalating global temperature, compounded by the increased intensification of production, heat stress significantly impacts the dairy industry.
Line 308-309 Give a reference for this sentence.
Line 316-317 The sentence needs to be reworded to make it clearer to your readers. Suggestion: In this study, experimental cows were undergoing heat stress (Summer: THI = 81.08 ± 4.57), triggering a series of thermoregulation.
Line 322-324 Consider rephrasing the sentence to make it easier for your readers to follow. Suggestion: It has been shown that cows' milk production in the heat-stressed group decreased by about 10-20% compared to the group under appropriate temperature conditions [27, 28], which is consistent with our results.
Line 334-335 the sentence’s structure could be improved upon to make it clearer to readers. Suggestion: Moreover, many reports have confirmed the role of circRNAs both in development and disease by analyzing their expression profiles [32, 33].
Line 341-343 There is a need to recast this sentence to make it clearer. Suggestion: According to GO and KEGG pathway analysis results, critical pathways related to the metabolic processes and autophagy in dairy cows under heat stress were identified.
Line 347-351 Please give reference(s) for your readers to follow.
Line 375 The phrase ‘in comparison with’ should be changed to ‘compared with’.
Line 388-390 This sentence requires recasting to make it clearer to your readers. Suggestion: Another interesting finding is that circRNA22544 (EZH2) expression significantly differed between Sum1 and Spr1.
Line 402-403 This sentence is incomplete; consider rephrasing it with another sentence.
Line 403-405 The sentence is unclear and needs to be reworded to improve its clarity. Suggestion: Moreover, heat stress negatively impacts feed intake, milk yield, and energy negative balance in early lactation [48].
4. Materials and Methods
The authors stated in lines 434-436 that the experimental animals were fed a total mixed ratio (TMR) ad lib. It is essential at this point to expand by highlighting the nutritive composition of this ration as stated by the standard for dairy cattle feeding. This should be tabulated. This will enable your readers to follow the discussion better and for this study to be replicated elsewhere.
Line 443-446 The sentence appears informal and hard to follow. Rewrite it to improve its clarity. Suggestion: What is more, the diet's nutrient composition remained consistent throughout the test period, and its nutritional level was in accordance with the standard for dairy cattle feeding (NY/T 34-2004).
Line 490 The word ‘correspondent’ doesn’t seem to fit this context. Consider replacing it with a more appropriate word. Suggestion: ‘corresponding’.
Line 510 The phrase ‘In pursuit of uncovering the’, appears unclear; consider changing it. Suggestion: You may use ‘To uncover’ instead.
Conclusions: This section is not mandatory in the IJMS, but conclusions can be added to the manuscript. Some readers read the abstract and rush to read the conclusions only in a paper; if these are well written, they are encouraged to read the whole paper. Think about adding it.
References
All the IMJS and MDPI guidelines on references have not been followed and adhered to. Authors should go through all of them and make the necessary adjustments accordingly.
Author Response
First of all, we deeply appreciate the time and effort reviewers spent reviewing this article, and for key suggestions/comments. Your suggestions helped us improve the article in a better way. We have studied the comments carefully and have made corrections/responded which we hope to meet with approval.
Point 1: Generally, the paper is well-written and backed by recent publications. However, I have highlighted areas where the paper could be improved upon under the different topics.
Abstract: The abstract should be a brief report of the research work. Many readers hardly read further than the abstract. They view the abstract as a short form of the paper. The silent points from the paper should be presented as a summary in the abstract. A precise statement about the methods is always needed. Most important is a concise summary of the results. The conclusion should put the work in perspective and entice readers to read the entire work.
In the light of the above, the first 4 sentences (Lines 13-17) should be discarded or incorporated into the introduction.
Line 13 The phrase ‘rise of global temperature’ may be wordy; for clarity, consider changing it to “global temperature rise’.
Line 14 The word ‘negative’ is often overused. Consider using a more specific synonym to improve the clarity of the sentence. Suggestion: you may use the word ‘damaging’.
Line 15 It appears that a pronoun is missing after behind. Suggestion: Add the pronoun ‘it’ after behind.
Introduction: Line 41 The preposition (of) used here is incorrect. Suggestion use the preposition, ‘in’ instead of ‘of’.
Line 58 There is a pronoun problem here. Suggestion: Use ‘their’ instead of ‘its’.
Line 59-61 The sentence appears long and difficult to follow; consider breaking it into two. Suggestion: Subsequent research tried to uncover the functional role of circRNAs in various species and tissues. A handful of circRNAs have been demonstrated to have functions in post-transcriptional regulation, acting as miRNAs sponge [9].
Line 72 ‘number of’ is unnecessary; consider removing it.
Line 73 ‘that’ is unnecessary; consider removing it.
Line 75-77 The sentence is unclear and difficult to follow. There is a need to recast it. Suggestion: Also, copious changed circRNAs were discovered in cow mammary gland tissues [15, 17], hypothalamic, and pituitary tissues [18] under heat stress.
Line 86-89 A knowledgeable audience might find this sentence difficult to understand, it may be necessary to break it into two. Suggestion: Liu et al. described the 86 expression patterns of mRNA and miRNA in the blood of the same individual cows under different conditions, from heat stress to thermal equilibrium [21]. They suggested individuals' influence on cows' response to heat stress.
Line 93-94 Your sentence may be unclear or hard to follow; consider recasting it. Suggestion: Here, we examined the changes in milk yield, rectal temperature, and respiratory rate of dairy cows under heat stress.
Response 1: Thank you for your precious time in pointing these out. We are sincerely grateful for all these detailed and valuable suggestions. They really help us a lot in improving the paper’s readability. We have carefully revised the manuscript according to your comments. All the changes are highlighted in the revised manuscript.
Point 2: Results:
Line 131 The word ‘showed’ is in the wrong tense; change it to the right form. Suggestion: use ‘shows’ instead.
Line 176-177 Your sentence may be unclear or hard to follow; consider recasting it. Suggestion: The current study presented only the top 50 GO terms with significance (P < 0.05) and the top 20 KEGG pathways.
Discussion:
Line 306-308 This sentence is unclear or hard to follow; consider recasting it. Suggestion: With escalating global temperature, compounded by the increased intensification of production, heat stress significantly impacts the dairy industry.
Line 308-309 Give a reference for this sentence.
Line 316-317 The sentence needs to be reworded to make it clearer to your readers. Suggestion: In this study, experimental cows were undergoing heat stress (Summer: THI = 81.08 ± 4.57), triggering a series of thermoregulation.
Line 322-324 Consider rephrasing the sentence to make it easier for your readers to follow. Suggestion: It has been shown that cows' milk production in the heat-stressed group decreased by about 10-20% compared to the group under appropriate temperature conditions [27, 28], which is consistent with our results.
Line 334-335 the sentence’s structure could be improved upon to make it clearer to readers. Suggestion: Moreover, many reports have confirmed the role of circRNAs both in development and disease by analyzing their expression profiles [32, 33].
Line 341-343 There is a need to recast this sentence to make it clearer. Suggestion: According to GO and KEGG pathway analysis results, critical pathways related to the metabolic processes and autophagy in dairy cows under heat stress were identified.
Line 347-351 Please give reference(s) for your readers to follow.
Line 375 The phrase ‘in comparison with’ should be changed to ‘compared with’.
Line 388-390 This sentence requires recasting to make it clearer to your readers. Suggestion: Another interesting finding is that circRNA22544 (EZH2) expression significantly differed between Sum1 and Spr1.
Line 402-403 This sentence is incomplete; consider rephrasing it with another sentence.
Line 403-405 The sentence is unclear and needs to be reworded to improve its clarity. Suggestion: Moreover, heat stress negatively impacts feed intake, milk yield, and energy negative balance in early lactation [48].
Response 2: Thank you for the suggestion. Every comment mentioned by the reviewers has been corrected very carefully in the manuscript. Relevant important and current citations have been added to the discussion part and where else needed in the manuscript. All the changes are highlighted in the revised manuscript.
Point 3: Materials and Methods:
The authors stated in lines 434-436 that the experimental animals were fed a total mixed ratio (TMR) ad lib. It is essential at this point to expand by highlighting the nutritive composition of this ration as stated by the standard for dairy cattle feeding. This should be tabulated. This will enable your readers to follow the discussion better and for this study to be replicated elsewhere.
Line 443-446 The sentence appears informal and hard to follow. Rewrite it to improve its clarity. Suggestion: What is more, the diet's nutrient composition remained consistent throughout the test period, and its nutritional level was in accordance with the standard for dairy cattle feeding (NY/T 34-2004).
Line 490 The word ‘correspondent’ doesn’t seem to fit this context. Consider replacing it with a more appropriate word. Suggestion: ‘corresponding’.
Line 510 The phrase ‘In pursuit of uncovering the’, appears unclear; consider changing it. Suggestion: You may use ‘To uncover’ instead.
Conclusions: This section is not mandatory in the IJMS, but conclusions can be added to the manuscript. Some readers read the abstract and rush to read the conclusions only in a paper; if these are well written, they are encouraged to read the whole paper. Think about adding it.
References: All the IMJS and MDPI guidelines on references have not been followed and adhered to. Authors should go through all of them and make the necessary adjustments accordingly.
Response 3: We are thankful for the reviewer’s suggestion. We have added the Composition and nutrient levels of basal diets for dairy cattle feeding (supplementary files 10) in the Materials and Methods section. Moreover, the Conclusion section have been added in the manuscript. We have downloaded the endnote format of the references from the IMJS website and corrected every editorial issue you mentioned in detail. Thank you again for your enlightening suggestion.
Supplementary files 10 Composition and nutrient levels of basal diets
Ingredient |
Dry matter basis (%) |
Corn grain |
28.70 |
Wheat bran |
5.30 |
Soybean meal |
6.40 |
Cottonseed meal |
8.00 |
Chinese wild rye |
25.00 |
Alfalfa hay |
25.00 |
Calcium hydrogen phosphate |
0.55 |
Mineral and vitamin premix 1 |
0.50 |
Salt |
0.55 |
Nutrient composition |
Dry matter basis (%) |
NEL (MJ·kg-1) 2 |
6.48 MJ·kg-1 |
Crude protein |
15.40 |
Neutral detergent fiber |
39.60 |
Acid detergent fiber |
21.70 |
Ca |
0.80 |
P |
0.46 |
Note: 1 Contained (per kilogram) vitamin A 9.5×105 IU, vitamin D 4×104 IU, vitamin E 7.5×103 IU, Fe 3×103 mg, Cu 2×103 mg/kg, Mn 2.5×103 mg, Zn 8×103 mg, Co 20 mg, I 2100 mg and Se 60 mg. 2 NEL is calculated based on the composition of raw materials. The rest are measured values.
Reviewer 3 Report
The authors present a study on key circRNAs from blood samples and pathways of circRNAs associated with heat stress in Holstein cows. Changes in milk yield, rectal temperature, and respiratory rate of experimental cows between heat stress (summer) and non-heat stress (spring) conditions were recorded in each 15 cows (Sum1 vs Spr1: same lactation stage, different individuals) and in further 15 cows (Spr2: same individual as in Spr1, but different lactation stage).
Table 1: to which comparison refers the P-value. Please clarify or add the a further column for both comparisons. In the text, you refer to Sum and Spr (Line 102) and to Spr1 and Sum1 (Line 109).
The authors may add a Conclusion section.
The authors discuss the consequences of the study design and in addition to this, they should conclude for further studies which design and numbers of animals should be chosen.
Maybe the authors give some more justification for the design chosen.
The issue of number of animals per experimental group and possible consequences for the study outcomes should be addressed, also in association with previous reports. This means how robust are the results.
Author Response
First of all, we deeply appreciate the time and effort reviewers spent reviewing this article, and for key suggestions/comments. Your suggestions helped us improve the article in a better way. We have studied the comments carefully and have made corrections/responded which we hope to meet with approval.
Point 1: Table 1: to which comparison refers the P-value. Please clarify or add a further column for both comparisons. In the text, you refer to Sum and Spr (Line 102) and to Spr1 and Sum1 (Line 109).
Response 1: Thank you for the suggestion. We have added an additional column in Table 1 to indicate the P-value for Sum and Spr and Supplemented a description of Spr and Sum in the text. All the changes are highlighted in the revised manuscript.
Point 2: The authors may add a Conclusion section.
Response 2: Thank you for the valuable suggestion. We totally agree with this comment. Conclusion section have been added in the manuscript. All the changes are highlighted in the revised manuscript.
Point 3: The authors discuss the consequences of the study design and in addition to this, they should conclude for further studies which design and numbers of animals should be chosen. Maybe the authors give some more justification for the design chosen. The issue of number of animals per experimental group and possible consequences for the study outcomes should be addressed, also in association with previous reports. This means how robust are the results.
Response 3: We are appreciative of the reviewer’s suggestion. Thank you very much for your appreciation of the work. Your valuable comments are highly appreciated. In the investigation of heat stress on cattle, with the increase of experimental samples, the analysis is statistically more robust and reliable, but, in the same time, more manpower and expenses will be required. In practice, generally more than three samples per group are selected for analysis. In this manner, we randomly selected 5 animals from each per group of 15 animals for RNA-seq. We have also added a summary of the experimental design suggestions in the Discussion section. It is important for researchers to choose an appropriate comparison design for heat stress study. When we are more concerned about the effects of heat stress on the cows’ lactation performance of cows, the impact on cow lactation cannot be ignored. In this case, Comparison 1 is probably a better choice. However, supposing that if we are more concerned about the effects of heat stress on the cow's organism, we should balance the impact of the different lactation stages on the cows. Moreover, the breed of dairy cattle is also relevant to the choice of experimental sample design. Holstein cows have fewer individual differences than local breeds and are a better choice for comparison 2. More detailed information can be found in the revised manuscript.
Round 2
Reviewer 2 Report
IJMS-2411011-Peer-review-v2.
General comments
I reviewed the manuscript, and the authors adequately implemented the areas of corrections and suggestions earlier suggested. I detected a few areas that should be addressed by the authors, as presented below:
· Line 19 The preposition ‘on’ is wrongly used. Suggestion: Use ‘into’ instead.
· Line 42 ‘about’ is missing between brought and by. It should be ….brought about by…
· Line 333 cow’s’, should rather be cow’s.
· Line 346 Delete the word ‘of’.
· Line 426 The modifiers in the noun phrase ‘energy negative balance’ are in the wrong order; consider changing the word order. Suggestion: negative energy balance.
· Line 435 ‘The’ is missing between from and above.
· The word ‘design’ should be in the plural form (designs).
· Line 438 Replace ‘are’ with ‘is’.
· Line 565 replace ‘consisted’ with ‘consisting’.
- I observed an anomaly in the citation of some references. For instance, in line 79 Wang et al. identified several circRNA-associated-ceRNA networks involved in milk fat metabolism under heat stress [15]. Ideally, the number in the square bracket should follow the citation immediately rather than putting it at the sentence's end. Suggestion: Wang et al. [15] identified several circRNA-associated-ceRNA networks involved in milk fat metabolism under heat stress. The authors should check similar citations as presented in the manuscript (Hao et al. line 68; Liu et al. lines 92, 417; Yang et al. line 364; Luo et al. line 428; and Zhang et al, line 431…. Check the manuscript for others) and correct them accordingly.
IJMS-2411011-Peer-review-v2.
General comments
I reviewed the manuscript, and the authors adequately implemented the areas of corrections and suggestions earlier suggested. I detected a few areas that should be addressed by the authors, as presented below:
· Line 19 The preposition ‘on’ is wrongly used. Suggestion: Use ‘into’ instead.
· Line 42 ‘about’ is missing between brought and by. It should be ….brought about by…
· Line 333 cow’s’, should rather be cow’s.
· Line 346 Delete the word ‘of’.
· Line 426 The modifiers in the noun phrase ‘energy negative balance’ are in the wrong order; consider changing the word order. Suggestion: negative energy balance.
· Line 435 ‘The’ is missing between from and above.
· The word ‘design’ should be in the plural form (designs).
· Line 438 Replace ‘are’ with ‘is’.
· Line 565 replace ‘consisted’ with ‘consisting’.
- I observed an anomaly in the citation of some references. For instance, in line 79 Wang et al. identified several circRNA-associated-ceRNA networks involved in milk fat metabolism under heat stress [15]. Ideally, the number in the square bracket should follow the citation immediately rather than putting it at the sentence's end. Suggestion: Wang et al. [15] identified several circRNA-associated-ceRNA networks involved in milk fat metabolism under heat stress. The authors should check similar citations as presented in the manuscript (Hao et al. line 68; Liu et al. lines 92, 417; Yang et al. line 364; Luo et al. line 428; and Zhang et al, line 431…. Check the manuscript for others) and correct them accordingly.
Author Response
First of all, we deeply appreciate the time and effort reviewers spent reviewing this article, and for key suggestions/comments. Your suggestions helped us improve the article in a better way. We have studied the comments carefully and have made corrections/responded which we hope to meet with approval.
Point 1: I reviewed the manuscript, and the authors adequately implemented the areas of corrections and suggestions earlier suggested. I detected a few areas that should be addressed by the authors, as presented below:
Line 19 The preposition ‘on’ is wrongly used. Suggestion: Use ‘into’ instead.
Line 42 ‘about’ is missing between brought and by. It should be ….brought about by…
Line 333 cow’s’, should rather be cow’s.
Line 346 Delete the word ‘of’.
Line 426 The modifiers in the noun phrase ‘energy negative balance’ are in the wrong order; consider changing the word order. Suggestion: negative energy balance.
Line 435 ‘The’ is missing between from and above.
The word ‘design’ should be in the plural form (designs).
Line 438 Replace ‘are’ with ‘is’.
Line 565 replace ‘consisted’ with ‘consisting’.
I observed an anomaly in the citation of some references. For instance, in line 79 Wang et al. identified several circRNA-associated-ceRNA networks involved in milk fat metabolism under heat stress [15]. Ideally, the number in the square bracket should follow the citation immediately rather than putting it at the sentence's end. Suggestion: Wang et al. [15] identified several circRNA-associated-ceRNA networks involved in milk fat metabolism under heat stress. The authors should check similar citations as presented in the manuscript (Hao et al. line 68; Liu et al. lines 92, 417; Yang et al. line 364; Luo et al. line 428; and Zhang et al, line 431…. Check the manuscript for others) and correct them accordingly.
Response 1: We are thankful for the reviewer’s suggestions. Thank you for your precious time in pointing these out. We have carefully corrected every editorial issue you mentioned in detail. All the changes are highlighted in the revised manuscript.
